# Exploring Biological Evidence of Radioprotective Effects and Critical Oxygen Thresholds in *Zeugodacus cucurbitae* (Diptera: Tephritidae)

**DOI:** 10.3390/insects16080825

**Published:** 2025-08-08

**Authors:** Qing-Ying Zhao, YongLin Ren, Yun-Long Ma, Ju-Peng Zhao, Xin Du, Simon J. McKirdy, Guo-Ping Zhan

**Affiliations:** 1Chinese Academy of Quality and Inspection & Testing, Beijing 100176, China; zhaoqy2021@126.com; 2Harry Butler Institute, Murdoch University, 90 South St., Murdoch, WA 6150, Australia; y.ren@murdoch.edu.au; 3College of Environmental and Life Sciences, Murdoch University, 90 South St., Murdoch, WA 6150, Australia; 4Chinese Academy of Inspection and Quarantine Center for Biosecurity Sanya, Sanya 572000, China; mylsfh@126.com; 5Guangzhou Customs Technology Center, Guangzhou Customs District, Guangzhou 510623, China; zhaojp2005@126.com

**Keywords:** irradiation, modified atmosphere, radioprotective effects, critical oxygen threshold, melon fly

## Abstract

Controlling insect pests in fresh fruits and vegetables is important for preventing their spread through international trade. Irradiation is increasingly applied to prevent insect development or reproduction, rather than to kill them directly. To keep fruits fresh longer, producers often use special packaging called Modified Atmosphere Packaging (MAP), which reduces the oxygen level around the fruit. However, when insects are exposed to irradiation in low-oxygen conditions, they may survive better, making the treatment less effective. In this study, we investigate how different oxygen levels affect the ability of X-rays to kill larvae of the melon fly, a serious pest of tropical and subtropical fruits and vegetables. We found that when the oxygen level drops below 4%, the larvae become more resistant to radiation. Based on the estimated probit-9 value, an additional radiation dose of 13 to 18 Gy is required under anoxia (0% oxygen level) conditions to offset the radioprotective effect. Our findings help improve phytosanitary security standards for using radiation in combination with MAP. This will allow fruits to store fresh longer and remain pest-free during export, benefiting both farmers and consumers.

## 1. Introduction

The melon fly, *Zeugodacus cucurbitae* (Coquillett) (Diptera: Tephritidae), is a highly invasive pest affecting numerous fruit species across parts of Africa, the Middle East, South and Southeast Asia, and North America [1]. The melon fly is a significant quarantine pest that attacks 125 host plants, including economic fruits and vegetables, such as passionfruit, peach, mango, cashew, carambola, citrus, bitter gourd, pepper, sweet pepper, and tomato [2,3]. It causes damage to young, green, and soft-skinned fruits causing yield losses from 30 to 80% [4,5]. Phytosanitary treatments, including fumigation [6], irradiation [7,8], and temperature treatment [9,10] have been reported to control the melon fly in various commodities.

Methyl bromide (MB) fumigation, a critical method for controlling quarantine pests, is being phased out under the Montreal Protocol and Copenhagen Amendment due to ozone depletion concerns [11]. However, MB is still employed for quarantine and pre-shipment applications due to the lack of viable alternatives [12,13]. Consequently, there is an urgent need to develop efficient, cost-effective, and environmentally friendly MB alternative phytosanitary treatments [14]. Phytosanitary irradiation (PI) treatments have been confirmed as one of the most promising alternatives to MB for controlling insect pests in fresh and stored commodities [15,16,17].

Fresh products can be irradiated in insect-proof packaging for disinfestation and decontamination [18]. Modified atmosphere (MA) packaging (MAP) is widely employed to preserve the quality and extend the shelf life of fresh fruits by reducing O_2_ levels (below 8%) and/or increasing carbon dioxide levels (above 10%) [19,20,21]. While high CO_2_ levels (above 35%) may eliminate insects, they may also degrade fruit quality by forming carbonic acid, leading to browning, skin damage, accelerated ripening, and off-flavors, particularly at higher concentrations [22,23,24]. Low O_2_ treatments with elevated nitrogen concentrations may control insect pests while preserving fruit quality, which generated growing interest in nitrogen as a viable alternative to CO_2_ [25,26]. However, the MA exhibits delaying effects, requiring longer exposure times. For instance, after 9 days of exposure to either 5% or 0.5% O_2_ atmospheres, the rates of pupariation for 3rd instar larvae of *Ceratitis capitata* (Wiedemann) were 95.0% and 3.7%, respectively. Correspondingly, rates for adult emergence were 41.0% and 0%, with adult mortality of 45.0% and 100%, respectively [27].

Irradiation can induce cell death in the presence of O_2_, a phenomenon known as the oxygen effect, explained by the oxygen fixation hypothesis [28]. This hypothesis posits that radiation creates non-repairable lesions in nuclear DNA, leading to cell death, particularly in the presence of diatomic oxygen [29,30]. Conversely, irradiation under hypoxic conditions can trigger a radioprotective effect on insects, increasing their radio-tolerance. Therefore, anoxic and hypoxic environments are restricted in PI treatments until 2021, although when the restriction began is unclear [31,32,33].

The presence of a critical oxygen threshold in radioprotection has been demonstrated in several studies. For example, Follett et al. (2013) observed that 1–4% O_2_ increased survivorship in adults from 14 to 25%, but oxygen levels 3–8% and 11–15% did not increase survivorship when third-instar larvae of the melon fly were irradiated at 50 Gy under MAP conditions [8,34].

The critical oxygen threshold for radioprotective effects in late third-instar *Bactrocera dorsalis* (Hendel) larvae has been identified as approximately 4% O_2_ [35], while in *Trichoplusia ni* (Hübner), radiation resistance increased significantly when larvae were irradiated under hypoxic conditions below the species’ critical partial pressure of 3.3% O_2_ [33,34]. Significant radioprotective effects were also observed in *Drosophila suzukii* (Matsumura) when larvae were exposed to 20 Gy under low oxygen levels (≤5%) [24]. In contrast, no enhanced radioprotective effects were found in *D. suzukii* irradiated at 60 Gy under hypoxic conditions ranging from 3.2% to 15.4% O_2_ [36]. These findings suggest that the critical oxygen threshold for effective radioprotection may vary among insect species and remains to be clearly defined.

In addition, Dias et al. (2020) conducted confirmatory tests showing that MA conditions do not compromise the efficacy of PI doses recommended for several fruit flies, including *Anastrepha fraterculus* (Wiedemann), *Anastrepha ludens* (Loew), *B. dorsalis*, and *C. capitata* [37]. This finding suggests that increasing the radiation dose may compensate for the radioprotective effects. Consequently, the IPPC has removed the restriction on using MAP for the PI treatment of fruit flies [31,38]. However, there is still a lack of generic critical oxygen threshold information for radioprotection and a maximum radiation dose to overcome radioprotective effects. Therefore, further investigation is more critical for non-tephritid pests.

The melon fly is reported as the most radio-tolerant species among oriental fruit flies and Mediterranean fruit flies, with a generic radiation dose of 150 Gy (under ambient air conditions) established by the IPPC in 2009 [7,31]. Late third-instar larvae of tephritids are used as the target stage in PI treatments, and this study aims to test the hypothesis that radioprotective effects occur only when the oxygen concentration falls below a critical threshold.

## 2. Materials and Methods

### 2.1. Insect Rearing

Progeny of the melon fly were collected from infested bitter gourds (*Momordica charantia* L.) in Guangzhou, China, in July 2022. The replacement population consisted of a field-collected population of the melon fly established within a year. Late third-instar larvae, either naturally emerging from or manually dissected out of infested bitter gourds collected from the field, were used to initiate the population. These larvae were transferred to moist sand for pupariation and subsequently reared under the controlled laboratory conditions (26 ± 1 °C, 50–70% RH, 14:10 h L:D photoperiod) of Phytosanitary Treatment and Equipment, the former Chinese Academy of Inspection and Quarantine, Beijing, China. This replacement ensured the continuity of the colony with a genetic background closely reflecting natural field conditions, minimizing potential laboratory adaptation effects.

One day before adult eclosion, the puparia were transferred to rearing cages (40 cm × 40 cm × 50 cm) for emergence. Adults were provided with water, fresh pumpkin (*Cucurbita maxima* Duch.), and a powdered feed mixture (sucrose: protein: peptone = 9:3:1) [39]. Insecticide-free pumpkin pulp slices (approximately 1 cm thick) were used for egg collection and larval development, with late larvae pupariating in moist sand until adult emergence. For the irradiation treatment, late larvae from the third to seventh generations were utilized.

### 2.2. Irradiation Treatments

Preparing third-instar larvae. Late third-instar larvae of the melon fly were obtained from pumpkin fruits, either through dissection or the collection of emerged individuals. Over 120 larvae were placed in a 200-mesh nylon mesh bag for treatment. During irradiation, 3–6 bags containing larvae were combined into a 2 L gastight bag (Dalian Delin Gas Packaging Co., Ltd., Dalian, China) and sealed. The air within the bag was thoroughly evacuated using a diaphragm pump, after which it was filled with either pure nitrogen (≥99.999%) or a target MA (1, 2, 3, 4, or 5% O_2_ balanced with nitrogen) (Beijing Green Oxygen Tiangang Technology Development Co., Ltd., Beijing, China), maintaining the atmosphere for one minute. To ensure gas purity, this exhausting and injecting cycle was repeated three times, following the procedure by Zhan et al. (2020) for irradiating *B. dorsalis* larvae in a low-oxygen environment [35]. For irradiation treatment in ambient air, only the third-instar larvae were wrapped and placed in the 2 L bags.

The bags containing larvae and approximately 1.5 L of MA gas were maintained at room temperature before irradiation, which was conducted within a controlled exposure time of 2 h. Oxygen levels were monitored using a Headspace Gas Analyser OXYBABY^®^ 6.0 (WITT-Gasetechnik, Witten, Germany) after 0 (initial time), 0.5, 1, 1.5, and 2 h. The results are presented as mean ± SD, and the experimental workflow is shown in Figure 1.

Irradiator. An RS-2000 Pro X-ray irradiator (Rad Source Technologies, Inc., Coral Springs, FL, USA) was used to conduct all the irradiation treatments of the melon fly under MAP. The operating parameters were set at 220 KV and 17.6 mA. To ensure uniform dose distribution, a patented reflector was positioned at the bottom of the exposure chamber (dimensions: width 43 cm, depth 38 cm, height 43 cm), achieving 95% uniformity [40]. The absorbed dose was monitored using a UNIDOS dosimeter (PTW, Breisgau, Germany). Dose verification testing was performed on the irradiator at the Aerial Accredited Tests Laboratory. In 2023, a dosimetry comparison by the laboratory confirmed absorbed doses of 100 and 396 Gy, corresponding closely to the reference values of 100 and 400 Gy.

Irradiation. The MAP bags containing the test insects were placed inside the chamber of the X-ray irradiator and subjected to doses of 0 (control), 16, 28, 40, 52, 64, 76, and 88 Gy, respectively. Each dose had 3 replicates. The dose rate during irradiation was approximately 9.6 Gy/min. Upon reaching the desired radiation dose, the irradiator was stopped, and the nylon-net packaged bags containing the larvae were removed.

Pupariation and adult emergence. The irradiated larvae and the controls were transferred to the moist sand in plastic boxes and placed in the rearing room for pupariation and eclosion. Under normal rearing conditions, untreated melon fly adults emerge 10–13 d after pupariation. Therefore, mortality, defined as the prevention of adult emergence from third-instar larvae, was assessed 3 weeks later to allow sufficient time for the emergence of irradiated larvae [41].

### 2.3. Statistical Analysis

To analyze the individual effects of the radiation dose (16–88 Gy with 12 Gy increment) and oxygen level (0, 1%, 2%, 3%, 4%, 5%, and 21%), as well as the interaction effects of dose × O_2_ level, mortality data (failure of adult emergence) for the melon fly was subjected to two-way Analysis of Variance (ANOVA) after correction with Abbott’s formula. Means were compared using Tukey’s multiple comparison tests [42,43]. Linear regression following analysis of covariance (ANCOVA) was also used to analyze the arcsine-transformed mortality data to estimate the minimum dose causing 100% mortality at different O_2_ levels [8,43].

Additionally, all data derived from treatments and controls were analyzed using probit analysis with the PoloPlus 2.0 program to estimate the minimum dose for preventing adult emergence at each O_2_ level via the probit model [44,45]. Data less than 100% and the lowest dose causing 100% mortality were used in the ANCOVA and probit analyses. Furthermore, to compare the significance of radiotolerance between treatments at different O_2_ levels, pair-wise comparison tests were conducted by calculating 95% confidence limits (CL) of the lethal dose ratios (LDR) at LD_90_, LD_95_, LD_99_, and LD_99.9968_ (the minimum lethal dose causing 90%, 95%, 99% or 99.9968% mortality at 95% confidence level). If the 95% CL excludes 1, then the LDx values are significantly different [35,46,47].

## 3. Results

### 3.1. Mortality of Irradiated Larvae Under Different O_2_ Levels

The mortality of the melon fly was assessed after exposure to irradiation under anoxia, hypoxia, and ambient atmosphere, and was significantly affected by radiation dose (F = 484.7, df = 6,146, *p <* 0.0001), O_2_ level (F = 3.3, df = 6,146, *p =* 0.0052), and the interaction between radiation dose and O_2_ level (F = 2.8, df = 36,98, *p <* 0.0001), as determined by two-way ANOVA of the dose-mortality data.

The adult emergence rate substantially decreased across all O_2_ level treatment groups as the radiation dose increased (Table 1). At a dose of 76 Gy, no adults emerged when exposed to 4% and 5% O_2_ or ambient air. In contrast, approximately 0.5%, 1.0%, 0.8%, and 0.7% of melon flies emerged when irradiated under 0%, 1%, 2%, and 3% O_2_ atmospheres, respectively, indicating a radioprotective effect (more radio-tolerant) in these modified atmospheres. At a radiation dose of 88 Gy, no adults emerged at any oxygen level. However, for other radiation doses, there was no consistent pattern in mortality across different oxygen levels (Table 1).

A significant interaction between radiation and oxygen levels typically indicates a positive radioprotective effect. There was a significant difference in mean mortality observed under 0% O_2_ (69.6%), 1% O_2_ (76.0%), and 2% O_2_ (77.1%), indicating a significant radioprotective effect under 0% O_2_. However, no significant difference was observed compared to the other oxygen level treatments. Nevertheless, alternative statistical methods are warranted to assess the significance of oxygen effects between treatments conducted in ambient air and those in low-oxygen ambient conditions. This method may include techniques such as linear regression following ANCOVA, probit analysis with lethal dose ratio test, or other approaches that are better suited to the specific characteristics of the data and experimental design.

### 3.2. Critical Oxygen Threshold for Predicting the Radioprotective Effect

#### 3.2.1. Linear Regression After ANCOVA

The ANCOVA was conducted by the arcsine transformation of the Abbott’s corrected mortality data to establish the linear regression relationship between radiation dose and mortality. The results revealed the significant treatment effects on radiation dose (F = 1637.2, df = 6,140, *p* < 0.0001), O_2_ level (F = 6.6, df = 6,140, *p <* 0.0001), and the interaction effect between dose and O_2_ level (F = 5.0, df = 36,127, *p* < 0.0001). Consequently, the minimum dose required for achieving 100% mortality was predicted using linear regression to compare the relative radiotolerance among treatments in both low-oxygen and ambient air environments. The results (Table 2) indicate that all the coefficients of determination (*R*^2^) values, except for treatments under the 2% O_2_ atmosphere, exceeded 0.90, with a maximum value of 0.9743, suggesting a good fit of the data using the arcsine-transformation linear regression model.

Regarding the estimated dose leading to 100% mortality of third-instar larvae of the melon fly, they were remarkably consistent around 72–73 Gy under 21%, 5%, and 4% O_2_ levels, but increased notably to 81–82.2 Gy when exposed to 3%, 2%, 1%, and 0% O_2_ levels. Particularly, the dose value exhibited a rapid increase from 72 to 81 Gy (a 12.5% increase) as the O_2_ level decreased from 4% to 3% (Table 2). These findings indicate that a 4% O_2_ level is likely the critical oxygen threshold for the radioprotective effect in late third-instar melon fly larvae. The sequence of radiotolerance appears to be as follows: 0% ≈ 1% ≈ 2% ≈ 3% > 4% ≈ 5% ≈ 21% O_2_ atmospheres.

#### 3.2.2. Probit Analysis of Dose-Mortality Data

Probit analyses using the probit model without dose transformation were conducted on the dose-mortality data of late third-instar melon fly larvae. The estimated LD_90_, LD_95,_ LD_99_, and LD_99.9968_ values are presented in Table 3, along with slope, intercept, heterogeneity, lethal doses, and 95% confidence limits. The low heterogeneity and narrow confidence intervals indicate a good fit to the data. However, since all the CLs for the LDs overlap, the lethal dose ratio tests (Table 3 and Table 4) were accordingly used to compare the significance of the estimated lethal doses. In the probit model, it was found that the estimated LD_90_, LD_95_, and LD_99_ under 0% oxygen level are significantly higher compared to values under other oxygen levels (except for LD_99_ under 2% O_2_) and ambient air, indicating a significant radioprotective effect from 0% oxygen level.

Notably, the estimated LD_99_ and LD_99.9968_ values decreased sharply as oxygen levels increased from 3 to 4% and remained relatively stable thereafter up to 21%. Additionally, the LD_99_ and LD_99.9968_ under severe hypoxia (1%, 2%, and 3% O_2_) were significantly higher than those under moderate hypoxia (4% and 5% O_2_) and ambient air, where non-significant lethal dose values were observed (Table 3). This suggests a gradual decrease in the radioprotective effects with increasing oxygen levels, with the effect disappearing at a 4% oxygen atmosphere. Moreover, melon fly larvae exposed to 0% oxygen and severe hypoxia (1%, 2%, 3% O_2_), as well as to moderate hypoxia (4% and 5% O_2_) and ambient air, exhibited very similar slopes (0.054–0.057 vs. 0.067–0.071) and intercepts, indicating similar levels of radiotolerance within each group (Table 3). Thus, the suggested sequence of radiotolerance for the melon fly is as follows: 0% > 1% ≈ 2% ≈ 3% > 4% ≈ 5% ≈ 21% O_2_ atmospheres, with the critical oxygen threshold for producing a radioprotective effect identified around 4% O_2_ atmosphere.

### 3.3. Comparison of Radioprotective Effects

This observation suggests that the 4% O_2_ level represents the critical oxygen threshold. The estimated dose-probit lines (Figure 2), generated using the probit model, are employed to explain the irregular sequence of lethal doses of 90%, 95%, 99%, and 99.9968% with a 95% confidence level when third-instar melon fly larvae were exposed to X-ray radiation under hypoxia and ambient air conditions.

It is evident that the dose-probit lines for 0%, 1%, and 3% O_2_ levels are parallel, as are those for 4% and 21% O_2_ levels, with the latter group exhibiting steeper slopes (0.057 vs. 0.067–0.068) as shown in Table 3. Additionally, the dose-probit lines can be clearly categorized into three groups, indicating significant differences in radiotolerance among anoxia, hypoxic conditions, and ambient air. Group 1 (0% O_2_) shows the lowest intercept (Table 3), indicating the highest level of radiotolerance. Group 2 (1%, 2%, and 3% O_2_) exhibits noticeable and consistent gaps compared to group 1 after LD_90_, although only gaps A and B are significant. Meanwhile, clear distinctions between group 2 and group 3 (4%, 5%, and 21% O_2_) appear after LD_99_, and these differences become more pronounced, particularly after LD_99.99_ (gap E). Therefore, radioprotective effects are evident at the phytosanitary dose (LD_99.99_ or LD_99.9968_). While the line for 2% O_2_ shows a smaller slope, resulting in reduced increments within group 1, and gaps C, D, and F are insignificant, a noticeable radioprotective effect is still evident for PI treatment in atmospheres with oxygen levels below 4%.

## 4. Discussion

Radiotolerance in insects increases with increasing age and the developmental stage. Consequently, the most developed stage of insects in commodities exhibits the highest radiotolerance for PI treatment [15,48,49,50]. Consequently, late third-instar melon fly larvae, representing the most-developed stage in infested commodities, were selected for X-ray irradiation under various levels of modified atmosphere. Identical doses (16–88 Gy with a 12 Gy increment) were applied to assess the biological response related to radioprotection, with the prevention of adult emergence serving as the criterion for efficacy. The results demonstrated that irradiation under anoxia (0% O_2_) and extreme hypoxia (1%, 2%, and 3% O_2_) does induce a radioprotective effect, with the effect becoming more pronounced as the oxygen level decreases. This finding was supported by analysis of the dose–response data using Two-way ANOVA, linear regression following ANCOVA, and probit analysis. Similar radioprotective effects have been reported in various insect species, including *Grapholita molesta* (Busck) [51], *C*. *capitata* [52], *Ostrinia nubilalis* (Hübner) [53], *Anastrepha suspensa* (Loew) [54], *Z*. *cucurbitae* [8], *T*. *ni* [33,34], *D*. *suzukii* [24], and *B*. *dorsalis* [35].

The radioprotective effects, particularly noticeable at lower irradiation doses, have been observed in atmospheres with oxygen concentrations below 5%. Significantly, the most robust radioprotective effects were observed at 0% oxygen level in late larvae of the melon fly, as illustrated in Table 3 and Figure 2. The 0% oxygen level triggers a potent stimulatory protective response in insects, known as low-oxygen hormesis (also referred to as cross-tolerance, preconditioning, or postconditioning). This mechanism may be associated with the oxidative stress preparation hypothesis [55,56,57].

However, the radioprotective effects diminished as oxygen content exceeded 4% (Table 1, Table 2 and Table 3, Figure 2), indicating the existence of a critical oxygen threshold. These findings align with previous research by Follett et al. (2013), which demonstrated increased survivorship to adult stage in melon fly larvae irradiated in Ziploc storage bags with 1–4% O_2_ [8,56,57]. The identified critical oxygen threshold value is similar to that observed in *B*. *dorsalis* (≈4%) and *T*. *ni*, where oxygen levels between 2.5 and 5% have shown similar effects [34,35].

Furthermore, to identify significant differences in radiotolerance, dose–response data require probit analysis [33,34,35]. Notably, no statistically significant difference in the radioprotective effects was observed below the LD_99_ (Table 3, Figure 2), even when employing LDR at LD_90_, LD_95_, LD_99_, and LD_99.9968_. The sensitivity of the LDR test to overlapping confidence limits necessitated careful application, complemented by ANOVA on the dose-mortality data [35,45,47]. Additionally, in the dose–response test conducted under ambient air, the minimum dose resulting in 100% mortality was 76 Gy, with estimates from linear regression ranging from 72 to 73 Gy (Table 2). This value is within the range of previous reports [7].

The radioprotective effects, however, become significant when the lethal dose exceeds 99.99% at a 95% confidence level (Figure 2), meeting the minimum efficacy requirement for certain countries [48,58]. Phytosanitary treatments consistently administer overdoses to surpass stringent efficacy standards, such as the LD_99.9968_ for fruit flies. In our understanding, this overdosing strategy is primarily intended to ensure that treatment parameters do not fall below the required levels, and it may also incidentally help counteract potential radioprotective effects observed under certain modified atmospheres, such as the additional ≤12.6 Gy estimated for the melon fly (Table 4) or ≤13.9 Gy for *B. dorsalis* [35]. Consequently, previous studies on PI treatments for fruit flies indicated that MA did not diminish the effectiveness of the recommended treatments [37]. This led the IPPC to remove related restrictions in 2021 [31,38,50].

In addition to *Z. cucurbitae*, the effects of low-oxygen environments on other quarantine pests, particularly surface feeders (mealybugs, thrips, and mites), deserve further investigation. Unlike internal feeders protected within host tissues, surface-dwelling pests are more directly exposed to oxygen fluctuations. These pests might exhibit stronger radioprotective responses under hypoxia due to their reliance on oxygen-mediated damage mechanisms during irradiation [14]. This may raise challenges for developing generic PI protocols across diverse pest groups.

Furthermore, some commodities, such as apples and pears, are commercially stored under low-oxygen and cold conditions to prolong shelf life. If irradiation is applied without reoxygenation, this could unintentionally reduce treatment efficacy [59]. Therefore, pre- and post-treatment atmosphere management should be considered when integrating PI into supply chains involving cold-chain MAP or MA storage.

Recent advancements have enhanced our understanding of how insects respond to low-oxygen environments, particularly in utilizing brief low-oxygen exposures for radiation protection. Ionizing radiation damages biological systems through direct and indirect effects, the latter involving the generation of reactive free radicals that modify fatty acids and proteins, thereby impacting insect physiology [60,61,62]. Studies suggest that hypoxia-induced reductions in mitochondrial activity decrease reactive oxygen species (ROS) production, potentially enhancing insect radiation resistance. Hypoxia also stimulates antioxidant activity, further lowering ROS levels and oxidative damage [55,63]. Additionally, hypoxia activates various stress response pathways, including the upregulation of the heat shock protein in insects exposed to ionizing radiation [54,64].

The complex physiological responses to stressors, including low oxygen, exposure duration, treatment methods, and low-dose radiation hormesis, likely contribute to the observed radioprotective effects. However, the mechanisms underlying low-oxygen hormesis, the critical oxygen threshold levels, and the inconsistent results observed under 1% oxygen (Table 4, Figure 2) remain unclear, highlighting the need for further research [57,65,66]. Such research could pave the way for the broader application of irradiation in phytosanitary treatment, food preservation, and life sciences, and also lead to understanding the underlying mechanisms involved.

## 5. Conclusions

This study indicates significant variability in radiotolerance at the 0% oxygen level, as evidenced by two-way ANOVA, linear regression, and probit analysis of the dose–response data. Probit analysis further reveals significant differences in radiotolerance across 1%, 2%, and 3% O_2_ levels, while insignificance is observed at 4% and 5% O_2_ atmosphere and ambient air. Consequently, the critical oxygen threshold for radioprotection in melon fly larvae is found to be at the 4% O_2_ level. Moreover, an estimated maximum radiation dose of 13–18 Gy is required to counteract this effect during phytosanitary irradiation treatment according to the probit analysis. Further exploration of the mechanisms underlying the radioprotective effect, including metabolomics, transcriptomics, and oxidase studies, is encouraged. Establishing a generic threshold for oxygen concentration could facilitate the application of PI in MAP for pests other than the fruit fly. This advancement may promote the adoption of PI technology in international trade, offering a viable alternative to MB fumigation and supporting food security economic growth and trade development.

## Figures and Tables

**Figure 1 insects-16-00825-f001:**
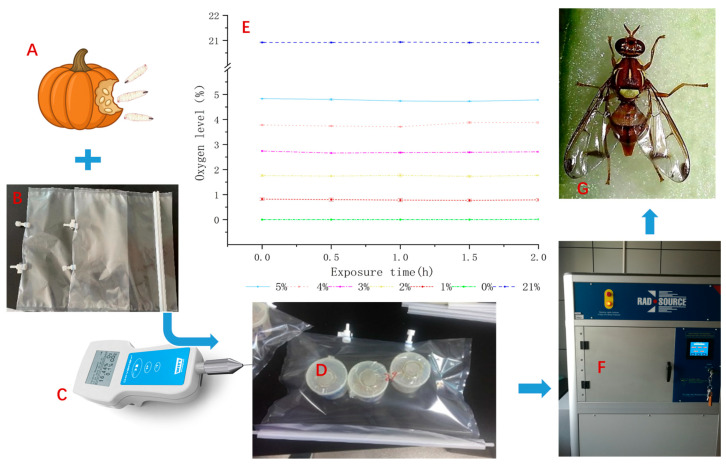
Experimental workflow and measured oxygen concentration inside MAP bags prior to irradiation. (**A**) Larva reared from pumpkin fruits; (**B**) Airtight treatment bag; (**C**) OXYBABY^®^ oxygen analyzer; (**D**) Larvae placed in airtight bag; (**E**) Oxygen concentration measured during the 2 h MA exposure; (**F**) X-ray irradiation; (**G**) Emerged adult of *Zeugodacus cucurbitae*.

**Figure 2 insects-16-00825-f002:**
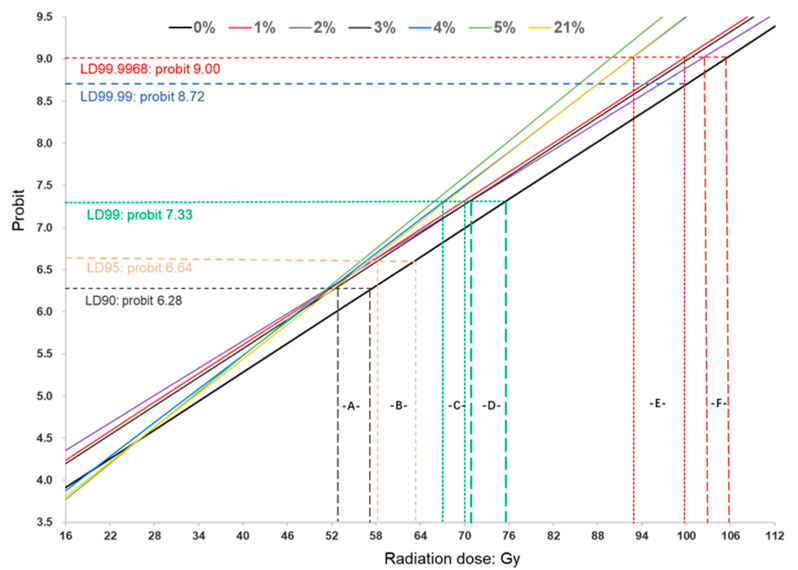
The estimated dose-probit lines for melon fly larvae irradiated under hypoxia and ambient air. A: minimal LD_90_ gap between group 1 (0% O_2_) vs. group 2 (1%, 2%, and 3% O_2_) and group 3 (4%, 5%, and 21% O_2_); B: minimal LD_95_ gaps among group 1 vs. group 2 and 3; C: minimal LD_99_ gaps between group 2 vs. group 3; D: minimal LD_99_ gaps between group 1 vs. group 2; E: minimal LD_99.9968_ gaps between group 2 vs. group 3; F: minimal LD_99.9968_ gaps between group 1 vs. group 2.

**Table 1 insects-16-00825-t001:** Radiation effects on the prevention of adult emergence in melon fly late larvae irradiated with X-rays under anoxia, hypoxia, and ambient air.

Oxygen Level	No. Larvae	Corrected Mortality of Irradiated Melon Fly at Different Radiation Doses (%, Mean ± SD)						
		16 Gy	28 Gy	40 Gy	52 Gy	64 Gy	76 Gy	88 Gy
0%	4573	18.3 ± 1.1	23.6 ± 1.1	70.6 ± 8.4	77.8 ± 7.9	97.2 ± 0.5	99.5 ± 0.2	100.0 ± 0.0
1%	3793	19.2 ± 2.6	54.8 ± 6.4	70.4 ± 9.0	90.3 ± 2.4	98.3 ± 0.1	99.0 ± 0.2	100.0 ± 0.0
2%	4338	17.1 ± 0.8	60.8 ± 3.4	81.4 ± 4.6	85.2 ± 6.5	97.0 ± 0.9	98.2 ± 0.7	100.0 ± 0.0
3%	5176	24.3 ± 0.3	46.1 ± 2.8	67.6 ± 3.3	89.3 ± 3.5	97.8 ± 0.7	99.3 ± 0.3	100.0 ± 0.0
4%	5470	18.6 ± 0.2	39.2 ± 4.3	57.7 ± 8.9	97.2 ± 0.5	98.8 ± 0.4	100.0 ± 0.0	100.0 ± 0.0
5%	3810	6.7 ± 2.0	49.0 ± 4.1	56.1 ± 5.9	93.3 ± 0.7	98.1 ± 0.8	100.0 ± 0.0	100.0 ± 0.0
21%	3572	6.0 ± 1.3	44.9 ± 4.9	60.4 ± 1.4	90.9 ± 1.2	98.2 ± 0.3	100.0 ± 0.0	100.0 ± 0.0

**Table 2 insects-16-00825-t002:** Results of linear regressions on dose-mortality data for melon fly late larvae irradiated under anoxia, hypoxia, and ambient air.

Oxygen Level	Regression Equation	Coefficient of Determination (*R*^2^)	Estimated Dose (Gy) for 100% Mortality
0%	y = 1.2314x − 10.5448	0.9077	81.6
1%	y = 1.0907x + 1.4373	0.9071	81.2
2%	y = 1.0373x + 4.7698	0.8797	82.2
3%	y = 1.1238x − 1.0476	0.9382	81.0
4%	y = 1.4091x − 11.3996	0.9458	72.0
5%	y = 1.4752x − 16.6165	0.9423	72.3
21%	y = 1.4713x − 17.3432	0.9743	73.0

**Table 3 insects-16-00825-t003:** Results of probit analysis on dose-mortality data for melon fly larvae irradiated under anoxia, hypoxia, and ambient air.

O_2_ (%)	No. Treated	Slope ± SE	Intercept ± SE	* Estimated Lethal Dose (Gy) and Its 95% Confidence Intervals				Heterogeneity
				LD_90_	LD_95_	LD_99_	** LD_99.9968_	
0	3184	0.057 ± 0.002	−1.998 ± 0.084	57.2 (52.9–63.0) a	63.6 (58.5–70.7) a	75.4 (68.7–85.3) a	104.6(93.3–121.6) a	8.21
1	2863	0.057 ± 0.002	−1.672 ± 0.082	51.4 (47.8–55.9) b	57.7 (53.5–63.3) b	69.6 (63.9–77.2) b	98.7 (89.1–111.8) ab	4.21
2	3111	0.054 ± 0.002	−1.506 ± 0.076	52.0 (47.2–58.7) b	58.8 (53.1–67.1) b	71.5 (63.8–83.3) ab	102.3 (89.5–123.5) a	8.84
3	3204	0.057 ± 0.002	−1.715 ± 0.081	52.6 (49.6–56.3) b	59.0 (55.4–63.5) b	70.9 (66.0–77.3) bc	100.3 (91.9–111.5) a	3.53
4	3909	0.067 ± 0.003	−2.194 ± 0.133	51.6 (48.7–55.5) b	57.0 (53.5–61.9) b	67.2 (62.3–74.1) cd	92.0 (83.4–104.7) bc	4.03
5	2366	0.071 ± 0.003	−2.366 ± 0.117	51.1 (48.3–54.8) b	56.2 (52.9–60.7) b	65.8 (61.3–71.9) d	89.3 (81.6–99.8) c	3.70
21	2691	0.068 ± 0.003	−2.281 ± 0.100	52.2 (49.4–55.8) b	57.6 (54.2–62.0) b	67.6 (63.0–73.6) cd	92.1 (84.4–102.5) bc	3.96

* Lethal dose value followed with different letters in the same column indicate significant differences (lethal dose ratio test, *p* < 0.05); ** LD_99.9968_ value was extrapolated from dose–response data.

**Table 4 insects-16-00825-t004:** Pairwise comparison of lethal dose ratio test for melon fly larvae irradiated under anoxia, hypoxia, and ambient air.

Reference O_2_	Pairwise O_2_	95% CL of LDR at LDx			
		LD_90_	LD_95_	LD_99_	LD_99.9968_
0%	1%	2.3 × 10^3^–2.0 × 10^8^	7.6 × 10^2^–6.7 × 10^8^	7.1 × 10^1^–8.4 × 10^9^	1.0 × 10^−2^–6.9 × 10^12^
	2%	6.6 × 10^2^–4.3 × 10^7^	7.4 × 10^1^–5.1 × 10^7^	9.2 × 10^−1^–9.3 × 10^7^	0.0–6.9 × 10^8^
	3%	2.0 × 10^2^–9.1 × 10^6^	6.0 × 10^1^–2.6 × 10^7^	4.7–2.4 × 10^8^	0.0–9.8 × 10^10^
	4%	2.0 × 10^3^–6.9 × 10^7^	5.7 × 10^3^–1.9 × 10^9^	2.9 × 10^4^–1.2 × 10^12^	8.7 × 10^5^–∞
	5%	6.5 × 10^3^–2.1 × 10^8^	4.0 × 10^4^–1.1 × 10^10^	2.3 × 10^4^–1.0 × 10^12^	1.3 × 10^9^–∞
	21%	5.7 × 10^2^–1.6 × 10^7^	2.1 × 10^3^–4.6 × 10^8^	1.8 × 10^4^–3.3 × 10^11^	2.3 × 10^6^–∞
1%	2%	1.0 × 10^−3^–8.7 × 10^1^	0.0–8.7 × 10^1^	0.0–2.1 × 10^2^	0.0–1.8 × 10^3^
	3%	0.0–1.9 × 10^1^	0.0–1.9 × 10^1^	0.0–5.7 × 10^2^	0.0–2.8 × 10^5^
	4%	2.0 × 10^−3^–1.4 × 10^2^	5.0 × 10^−3^–1.4 × 10^2^	2.1 × 10^−2^–2.9 × 10^6^	3.5 × 10^−1^–5.6 × 10^13^
	5%	7.0 × 10^−3^–4.5 × 10^2^	3.6 × 10^−2^–4.5 × 10^2^	6.5 × 10^−1^–5.2 × 10^7^	4.8 × 10^2^–∞
	21%	1.0 × 10^−3^–3.3 × 10^1^	2.0 × 10^−3^–3.3 × 10^1^	1.3 × 10^−2^–8.0 × 10^5^	8.6 × 10^−1^–1.6 × 10^13^
2%	3%	1.0 × 10^−3^–6.7 × 10^1^	1.0 × 10^−3^–5.8 × 10^2^	0.0–4.4 × 10^4^	0.0–3.0 × 10^9^
	4%	1.0 × 10^−2^–5.1 × 10^2^	6.8 × 10^−2^–4.2 × 10^4^	1.9–2.2 × 10^8^	3.5 × 10^3^–∞
	5%	3.1 × 10^−2^–1.6 × 10^3^	4.7 × 10^−1^–2.4 × 10^5^	5.9 × 10^1^–4.0 × 10^9^	4.8 × 10^6^–∞
	21%	3.0 × 10^−3^–1.2 × 10^2^	2.4 × 10^−3^–1.0 × 10^4^	1.1–6.1 × 10^7^	8.6 × 10^3^–4.0 × 10^9^
3%	4%	4.5 × 10^−2^–1.7 × 10^3^	1.3 × 10^−1^–5.1 × 10^4^	7.1 × 10^−1^–4.4 × 10^7^	2.4 × 10^1^–1.3 × 10^15^
	5%	1.5 × 10^−1^–5.2 × 10^3^	9.2 × 10^−1^–3.0 × 10^5^	2.2 × 10^1^–7.8 × 10^8^	3.4 × 10^4^–∞
	21%	1.3 × 10^−2^–3.8 × 10^2^	4.8 × 10^−2^–1.3 × 10^4^	4.3 × 10^−1^–1.2 × 10^7^	6.2 × 10^1^–3.7 × 10^14^
4%	5%	1.9 × 10^−2^–5.2 × 10^2^	1.3 × 10^−2^–3.1 × 10^3^	4.0 × 10^−3^–1.3 × 10^5^	0.0–2.0 × 10^9^
	21%	2.0 × 10^3^–3.8 × 10^1^	1.0 × 10^−3^–1.3 × 10^2^	0.0–1.9 × 10^3^	0.0–2.4 × 10^6^
5%	21%	1.0 × 10^−3^–1.2 × 10^1^	0.0–1.9 × 10^1^	0.0–6.0 × 10^1^	0.0–1.6 × 10^3^

## Data Availability

The original contributions presented in this study are included in the article. Further inquiries can be directed to the corresponding authors.

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
