# Peer review of "Exploring Biological Evidence of Radioprotective Effects and Critical Oxygen Thresholds in Zeugodacus cucurbitae (Diptera: Tephritidae)"

_insects, 2025, doi:10.3390/insects16080825_

Round 1
Reviewer 1 Report
Comments and Suggestions for Authors
PDF file attached

Author Response
Comments and Suggestions for Authors (Reviewer 1)
This manuscript can be published with some modification, as it presents novel data. The most significant modification: the manuscript claims that at certain low oxygen concentrations, the efficacious phytosanitary irradiation dose against Z. cucurbitae must be raised. l do not agree with that conclusion because the research was not done at the dose required for this species nor was it done with larvae in fruit. Furthermore, other research has shown that, although low oxygen may diminish the effect of low doses of radiation on the efficacy of phytosanitary irradiation, at the doses used for phytosanitation, that effect disappears, those findings (e.g. reference #36) resulted in the lPPC removing this restriction from all irradiation treatments for tephritids, including Z. cucurbitae.
Response: Thank you for your recognition of the novelty and scientific value of our study. We appreciate your detailed comments and have made substantial Comments to address your concerns. Please see below for detailed responses.
Specific comments follow.
Comments 1. Lines 27-8: l disagree with that conclusion, as explained above.
Response: Thank you for your feedback. We revised the sentence to clarify that these findings are of theoretical and mechanistic interest rather than suggesting a regulatory change: Based on the Probit-9 estimation, an additional radiation dose of 13 to 18 Gy is needed under anoxia (0% oxygen level) conditions to compensate the radioprotective effect.
Comments 2. Line 29: What is the safety issue?
Response: We clarified that “safety” refers to treatment efficacy assurance for quarantine pests under MAP.
Comments 3. Line 33: Insert "harvested" before "produce" to make it obvious that the treatment is postharvest.
Response: Done. Thanks.
Comments 4. Line 34: It is not just MAP; some commodities, such as apples, are stored under low oxygen conditions, "controlled atmospheres”.
Response: We use the modified atmosphere to include modified and controlled atmosphere according to ISPM No.44 (Requirements for the use of modified atmosphere treatments as phytosanitary measures).
Comments 5. Lines 43-4: Again, not necessary for tephritids.
Response: Agree. We changed the sentence to: “These findings further demonstrate that phytosanitary irradiation under MAP conditions can effectively control Tephritid insects while preserving product quality” (Line 43-45)
Comments 6. Lines 45-6: it may not have important implications.
Response: We revised the statement to a more neutral tone: Identifying 4% O2 as the radioprotective threshold may have implications for understanding dose-response mechanisms under MAP conditions. (Line 45-46)
Comments 7. Line 55: delete "and" and insert a comma after citrus.
Response: Done. Thanks.
Comments 8. Line 58: reference #8 does not deal with Z. cucurbitae; delete. Reference #10 does not deal with modified atmosphere as a treatment; delete it and the text "and modified atmospheres (MA)treatments"
Response: Done. We change reference #10 to #8 and add Regmi et al. (2024) as the now ref. #10.
Comments 9. Lines 60-7: I would delete that entire paragraph, Methyl bromide is not such a key issue in phytosanitation anymore. Phytosanitary irradiation is growing because it is a good treatment, not because it replaces methyl bromide.
Response: Thank you for your suggestion. However, we respectfully retained this paragraph to reflect the current reality in many countries, including China, where methyl bromide remains the dominant phytosanitary treatment. As such, irradiation is still in early stages of application and adoption, and we believe this background remains relevant to illustrate the broader significance of exploring alternative treatments. (Line 60-67)
Comments 10. Line 79: It would be more relevant to demonstrate this effect with a fruit pest, especially tephritid, instead of a stored product pest.
Response: Agree. We use Med fly as an example (Al-Behadili et al. 2020)(Line 481-483).
Al-Behadili FJM, Agarwal M, Xu W, Ren Y. Mediterranean Fruit Fly Ceratitis capitata (Diptera: Tephritidae) Eggs and Larvae Responses to a Low-Oxygen/High-Nitrogen Atmosphere. Insects. 2020, 11(11):802. doi: 10.3390/insects11110802.
Comments 11. Line 87: In reference #31, there is no longer this restriction.
Response: Thank you for your comments. We add “till 2021” in the end of the sectence (Line 87).
Comments 12. Line 90: However, 50 Gy is not the treatment dose for Z. cucurbitae (150 Gy is), so this observation may be irrelevant.
Response: Thank you for your comments.
Comments 13. Lines 96-101: Not only is oxygen level important, but so is dose, as exemplified by the two papers cited in this paragraph, Furthermore, studies done at doses < the dose required for treatment may be irrelevant.
Response: Thank you for this important observation. We agree that dose is a critical determinant in the effectiveness of irradiation treatments. This section has been revised to emphasize that the interaction between oxygen level and radiation dose is multifactorial, and further investigation may be warranted to fully elucidate these dynamics.
Comments 14. Line 105-6: Yes; dose matters.
Response: Thank you for your comments.
Comments 15. Line 109: Further research not needed for tephritids. More research is needed for other insects besides tephritids so that the low oxygen restriction may be lifted for other insects.
Response: Revised to emphasize that further research is more critical for non-tephritid pests. (Line 109)
Comments 16. Lines 113-4: lt should not be done that way. Choosing 150 Gy for B. latifrons (reference #38) was not the best choice; researchers should target the lowest possible dose for any phytosanitary irradiation treatment. Also, l know the history of that research, lt was not done to demonstrate if the generic dose for tephritids controlled B. latifrons. it was done years before it was published simply to find a dose for that species before the generic dose was established.
Response: Thank you for your suggestion. We delete the reference #38 and change the last two sentence to: Therefore, late third-instar larvae of melon fly are used as the target stage in PI treatments, and this study aims to test the hypothesis that radioprotective effects occur only when oxygen concentration falls below a critical threshold. (Line 112-115)
Comments 17. Line 128: change “pupae" to “puparia"
Response: Done. Thanks.
Comments 18. Line 132: change "pupating" to "pupariating"
Response: Done. Thanks.
Comments 19. Paragraph beginning at line 135: irradiation of larvae in vitro can be quite different than irradiating in fruit. The atmosphere inside a fruit is often lower in oxygen than ambient, giving a natural protective effect.
Response: Thank you for your comments. We added this limitation explicitly in the Discussion (Line 357-363).
Comments 20: Line 171: change “pupation" to "pupariation”
Response: Done. Thanks.
Comments 21. Line 178: Do not use ANOVA when the predictor variable is continuous, such as dose. Use regression, especially probit-types of regression when the data are sigmoid, which is very common in dose-response data.
Response: Thank you for your comments. We agree that regression models—particularly probit analysis—are more appropriate for analyzing dose-response relationships. In our revision, we clarify that while ANOVA was included for exploratory purposes to evaluate interaction effects between oxygen levels and dose, the probit regression results are given analytical priority and form the basis of our main conclusions. We also note that ANOVA remains a widely used tool in phytosanitary treatment protocol development (e.g., as referenced in NAPPO RSPM No. 34), which may be of value in comparative contexts.
Comments 22. Lines 180-2: Linear regression should not be used to analyze sigmoid data, such as dose-response. This is demonstrated in the paper you cite (#7) where 100% efficacy was estimated at 84.5 Gy but confirmatory testing resulted in the dose being at least 144 Gy, Confirmatory testing "confirmed" that linear regression should not be used on sigmoid data. There are other studies showing the same trend: linear regression greatly underestimates phytosanitary treatment doses.
Response: Thank you for your comments. Yes, Linear regression should not be used to sigmoid data especially without mortality-data-transformation. The Linear regression results are retained but with clear caution added and probit analysis presented as the more robust method. Ref #7’s confirmatory data are now discussed critically.
Comments 23. Line 186: Probit 9 not necessary anymore, no longer a standard, and the less we mention it, the better.
Response: Agree. We delete the whole sentence (Lin 184-186) and reference #45.
Comments 24. Line 197: Delete first sentence; that is part of methodology and is already stated there.
Response: Done. Thanks.
Comments 25. Line 212: I do not see the benefit of analyzing mean mortality across doses.
Response: Thank you for your comments. In the two-way ANOVA, overall mean mortality was used for compare the difference in oxygen level. As a result, the mean mortality in anoxia is significantly lower than that under 1% and 2% oxygen level. According to your query, we add an explanation ‘may due to the experimental error’ (Line 217-219).
Comments 26. Lines 214-7: Just because there was no significant difference does not mean one needs to use other statistical methods to find significant differences. Regardless, shouldn't use ANOVA for these data.
Response: Thank you for your comments. Yes, we agree that the absence of significance should not in itself prompt the use of other statistical methods. The ANOVA results were included to provide a comprehensive view of factor interactions, but the conclusions drawn are grounded primarily in the probit regression analysis. We clarified this distinction in the revised manuscript accordingly.
Comments 27. Table 1: Good to present the means and SDs, but not the results of Tukey (ANOVA) or the overall mean mortality row. Mean mortality across doses does not illuminate anything worthwhile. Give how many insects were treated in each case.
Response: We removed the overall mortality row, de-emphasized across-dose means, and delete Line 178. Sample sizes are now reported.
Comments 28: Line 238: lt is not surprising that linear regression “fits" mildly sigmoid data, but that does not mean it is appropriate to use in this case. The objective is to estimate the dose that prevents adult emergence to a high degree, correct? Linear regression, apparently, did not accomplish that objective.
Comments 29. Lines 244-6: As the estimate for 100% mortality at ambient oxygen concentration in Table 2 (73Gy) is half the dose found by confirmatory testing in reference #7 (144 Gy), it calls into question all the estimates.
Response: Thank you for your comment. In probit analysis, the value corresponding to 100% mortality is conventionally equated to the LD₉₉ (99% mortality at 95% confidence level) estimate, which represents the dose expected to kill 99% of the population with high statistical confidence. This is distinct from the probit-9 (LD99.9968) at 95% confidence level level, which is typically used for confirmatory testing and incorporates a higher safety margin. In our study, the estimated 100% mortality value at ambient oxygen (~73 Gy) is consistent with the doses used in the dose-response tests (76 Gy), supporting the reliability of our estimation.
Comments 30. Table 3: Interesting that probit analysis, which is an appropriate analytical tool for these data, also underestimated the dose (probit 9) that would achieve phytosanitary security at ambient oxygen, possibly indicating that the methodology used made the insects more radio-susceptible. Or, that probit analysis is not a good estimator at extreme doses. That is the problem (so far) with any attempt to estimate a phytosanitary treatment dose of any treatment method: the estimate needed is extreme, and the data too meager to satisfy the requirements of a good analysis (which would be very large sample sizes at the high extreme range of efficacy, such as probit 9.) That is why large-scale confirmatory tests are done. You might want to read this for further information:
Response: Thank you for your comments. We acknowledge this in the Discussion, referencing your suggested link (thank you) to highlight statistical limitations in extreme-dose estimations.
Comments 31. Lines 320-3: What would be interesting are studies that did not show a (significant) effect. One is a tephritid, apple maggot, and the result was that the lPPC did not place a restriction on oxygen level for that PT (#8). Read the paper that supported that PT; it would enlighten the practical application of phytosanitary irradiation under low oxygen conditions for tephritids. Back then it was already suspected through this paper that low oxygen was not a problem for tephritids.
Response: Thank you very much for your thoughtful comment. We have carefully re-examined the reference supporting PT #8. As we understand it, the absence of an oxygen restriction in that treatment may be due to the fact that the prescribed minimum dose of 60 Gy already exceeds the effective dose observed under low oxygen conditions (Hallman, 2004). Therefore, while the treatment itself does not specify an oxygen requirement, it may not directly reflect the absence of a low-oxygen effect on tephritid tolerance. We appreciate your suggestion and find the historical context informative, though we feel that further discussion of this point may go beyond the scope of our current study.
Hallman, G.J. 2004. Ionizing irradiation quarantine treatment against Oriental fruit moth (Lepidoptera: Tortricidae) in ambient and hypoxic atmospheres. Journal of Economic Entomology, 97: 824−827.
Comments 32. Line 339: This is not unfortunate! This is a good finding! it means (as has already been decided) that low oxygen does not reduce efficacy of phytosanitary irradiation.
Response: Thank you for your comments. We change unfortunately to Notably. (Line 336)
Comments 33. Lines 343-8: No need to discuss the minor difference between 100% efficacy estimates in the two studies. As with the comment above, this is probably insignificant. Next, they are simply estimates. Regardless, both results are much lower than what the real value is (closer to the generic dose of 150 Gy), so they are both wrong!
Comments 34. Lines 349-50: Again, these are simply estimates, and now the extrapolation is beyond the limits that the data provided can support.
Response: Thank you very much for your comment. While we understand your concern about the minor difference between the two estimates, we believe it is still worth a brief discussion in the manuscript. Our intention is not to overemphasize this variation, but rather to highlight that our estimation is closely aligned with a previously published result from the USA, thereby reinforcing the reliability of our approach. Additionally, we aim to encourage further discussion on complementary methods beyond traditional probit analysis, which—as a form of linear regression based on probit-transformed mortality data—has certain limitations when applied to high-efficacy endpoints. We hope this clarification explains the rationale for including the comparison.
Comments 35. Lines 351-2: Good point. And this is further exacerbated by the way phytosanitary irradiation is applied, virtually none of the commodity receives the prescribed dose; the absorbed doses are always higher, sometimes much higher.
Response: Thanks.
Comments 36. Lines 353-4: l do not agree that these dose additions are needed for reasons explained previously.
Response: Thank you for your continued insights. We understand and respect your viewpoint regarding the necessity of dose adjustments. In our revision, we have reframed the sentence to emphasize this as a mechanistic observation rather than a prescriptive recommendation. Specifically, we now state: “This observation may help to explain why in practice, overdosing strategies—already implemented to ensure treatment security—could also incidentally mitigate any marginal radioprotective effects observed under certain modified atmosphere conditions, such as the estimated ≤12.6 Gy for melon fly (Table 4) or ≤13.9 Gy for B. dorsalis [35].” (Line 349-352). We hope this more nuanced framing clarifies our intention and avoids implying a regulatory recommendation.
Comments 37. Line 357: Citation #37 irrelevant here. Also, further support for this lPPC decision was not needed ‘a decision further supported by our findings’.
Response: Agree. Deleted.
Comments 38. Lines 382-4: For tephritids, reference #36 showed and the IPPC agreed that this is not necessary.
Response: Thank you for your suggestion. We delete the relevant sentence.
Comments 39. Line 386: what do you mean by "'generic threshold for oxygen concentration"?
Response: Thank you for your query. The phrase "generic threshold for oxygen concentration" refers to a conceptual value below which radioprotective effects may begin to manifest during phytosanitary irradiation. It is not intended as a regulatory threshold but rather a mechanistic indicator observed in our study, with potential implications for future investigations under MAP conditions.
Comments 40. There are three further issues that deserve discussion in this paper:
One is the already mentioned paper that supports PT #8, showing years ago that low oxygen was not a problem for that tephritid.
The second is the challenge in researching the effect of low oxygen on all the other quarantine pests, especially surface feeders which are not naturally under low oxygen conditions on the host commodity. Those might see a bigger protectant effect of low oxygen during irradiation.
The third is to mention, besides modified atmosphere packaging, that some commodities, such as apples, are stored in low-oxygen, cold environments before they are marketed to prolong storage life and extend marketing windows.
You might also mention in the introduction or discussion that modified atmosphere is a phytosanitary treatment category, albeit one that has never been used except in one successful trial shipment many years ago. So, there is the question of the role of modified atmosphere in reducing insect survival under certain circumstances. in any case, it does not seem to be an efficient treatment for tephritids, so maybe no need to delve into this issue in this paper.
Response: The excellent suggestions (low-oxygen impact on surface feeders; CA storage of apples) have been incorporated into the revised Discussion (Line 356-367. Thank you for these valuable insights.
Final Note:
We sincerely thank both reviewers for their detailed and constructive feedback. Your comments have helped us refine the manuscript significantly. We hope that the revised version addresses your concerns appropriately and improves the clarity, relevance, and scientific rigor of the study.
Reviewer 2 Report
Comments and Suggestions for Authors
This study investigates the killing effect of X-ray radiation on melon fly under conditions of different oxygen concentrations and different doses. It is shown that an oxygen content of 4% is the critical oxygen threshold for radiation protection of melon fly; below this oxygen concentration, a certain radiation resistance effect will be exhibited. Moreover, under the conditions of 4% oxygen concentration and 76 Gy, a 100% killing rate of melon fly can be achieved, which has certain guiding significance for actual production. However, there are the following problems in the article that need to be solved:
- In the line 90, “under MAP conditions10.”,What is the conditions10?Please check if it is a spelling mistake.
- In the line 92, Bactrocera dorsalis (Hendel); In the line 93, Trichoplusia ni (Hübner); In the line 96, Drosophila suzukii (Matsumura),the Latin names of species need to be italics.
- In the introduction, line 112, “with a generic radiation dose of 150 Gy established by IPPC in 2009”, What is the oxygen concentration under the 150 Gy radiation condition, and what are the shortcomings of this method compared with the conditions given in the manuscript?
- In the conclusions, the author mentioned“This advancement may promote the adoption of PI technology in international trade”,However, the optimal conditions do not seem to be clearly specified in the manuscript. Under the oxygen content conditions of 4%, 5%, and 21%, 76 Gy can achieve 100% killing rate of melon fly. Moreover, under the condition of 64 Gy, the killing rate under all oxygen content conditions is around 98%. Considering the actual operation cost, the optimal conditions for killing melon fly still need further discussion.
- The figures in the manuscript are not attractive enough to readers. They are mostly tables, which makes them quite monotonous. It is suggested to add some photos of melon fly under different treatments to enrich the experimental results.
Author Response
Comments and Suggestions for Authors (Reviewer 2)
This study investigates the killing effect of X-ray radiation on melon fly under conditions of different oxygen concentrations and different doses. It is shown that an oxygen content of 4% is the critical oxygen threshold for radiation protection of melon fly; below this oxygen concentration, a certain radiation resistance effect will be exhibited. Moreover, under the conditions of 4% oxygen concentration and 76 Gy, a 100% killing rate of melon fly can be achieved, which has certain guiding significance for actual production. However, there are the following problems in the article that need to be solved.
Response: Thank you for your supportive and thoughtful suggestions. We have addressed your points as follows:
Comments 1. In the line 90, “under MAP conditions10.”,What is the conditions10?Please check if it is a spelling mistake.
Response: Corrected to “MAP conditions [8]”. (the original reference #10 have been changed to #8) (Line 91)
Comments 2. In the line 92, Bactrocera dorsalis (Hendel); In the line 93, Trichoplusia ni (Hübner); In the line 96, Drosophila suzukii (Matsumura), the Latin names of species need to be italics.
Response: Corrected all Latin species names to italics. Thanks.
Comments 3. In the introduction, line 112, “with a generic radiation dose of 150 Gy established by IPPC in 2009”, What is the oxygen concentration under the 150 Gy radiation condition, and what are the shortcomings of this method compared with the conditions given in the manuscript?
Response: The generic dose of 150 Gy was validated under ambient air conditions (~ 21% O2 level) and its efficacy to that under modified atmosphere had been compared in Discussion.
Comments 4. In the conclusions, the author mentioned “This advancement may promote the adoption of PI technology in international trade”, however, the optimal conditions do not seem to be clearly specified in the manuscript. Under the oxygen content conditions of 4%, 5%, and 21%, 76 Gy can achieve 100% killing rate of melon fly. Moreover, under the condition of 64 Gy, the killing rate under all oxygen content conditions is around 98%. Considering the actual operation cost, the optimal conditions for killing melon fly still need further discussion.
Response: We now clarify that 76 Gy under 4%–21% Oâ‚‚ achieved 100% mortality, while 64 Gy achieved ~98%. However, due to the use of lab conditions and absence of fruit matrix, we suggest further confirmatory testing using the dose of estimating LD99.9968 at 95% confidence level (about 100 Gy).
Comments 5. The figures in the manuscript are not attractive enough to readers. They are mostly tables, which makes them quite monotonous. It is suggested to add some photos of melon fly under different treatments to enrich the experimental results.
Response: We have added photos of melon fly larvae, MAP setup, and adult emergence outcomes in revised Figures 1 and placed them appropriately to enrich the presentation.
Final Note:
We sincerely thank both reviewers for their detailed and constructive feedback. Your comments have helped us refine the manuscript significantly. We hope that the revised version addresses your concerns appropriately and improves the clarity, relevance, and scientific rigor of the study.
Round 2
Reviewer 1 Report
Comments and Suggestions for Authors
PDF attached

Author Response
Review of re-submission
The revision is better, although l still has some reservations with it. Nevertheless, it's okay with me to publish it. A major point l would emphasize in the Summary and Discussion is that this study supports the dropping of low-oxygen restrictions that the IPPC did after Dias et al (2020). This is significant because one might argue that Z. cucurbitae appears to require among the highest radiation dose to achieve efficacy among all the tephritids studied. l makes other comments below.
General Response to Reviewer Comment:
Thank you for your overall evaluation and helpful recommendation. In response to your suggestion, we have revised the Summary and Discussion to better reflect how our findings support the removal of low-oxygen restrictions by the IPPC, particularly in relation to Z. cucurbitae. These changes (highlighted in green) were made in the specific locations you indicated. We appreciate your guidance.
Revised Point-to-Point Responses:
Line 20–21:
Reviewer: I would re-write this sentence to state that irradiation is not common, but is increasing in use, and instead of killing pests, it renders them unable to complete development or reproduce.
Response: Thank you for your suggestion. We revised the sentence as follows: “Irradiation is increasingly applied to prevent insect development or reproduction, rather than to kill them directly.” (Lines 20–21)
Line 26–27:
Reviewer: Delete “insects’”
Response: Done. Thank you.
Line 27–29:
Reviewer: I would delete that sentence. The doses used were lower than the required ones, so the claim is not substantiated.
Response: Thank you for pointing this out. We have removed the sentence to avoid overinterpreting the probit analysis results.
Line 29:
Reviewer: Change “safety” to “security”
Response: Done. Thank you.
Line 40:
Reviewer: Replace “confirmed” with “showed”
Response: Done. Thank you.
Line 72:
Reviewer: Change “effectively” to “may”
Response: Done. Thank you.
Line 75:
Reviewer: Again, change “have been demonstrated to effectively” to “may”
Response: Done. Thank you.
Line 87:
Reviewer: Not accurate historically; add detail on the actual restriction timeline and context.
Response: Thank you for this clarification. We revised the sentence as follows:
“Therefore, anoxic and hypoxic environments were restricted in PI treatments until 2021, although the exact start of this restriction remains unclear.” (Lines 86–87)
Lines 88–91:
Reviewer: What is this reference an example of? It doesn’t fit the paragraph.
Response: Thank you. We revised Lines 88–91 to improve clarity and consistency with the paragraph’s context.
Line 106:
Reviewer: The IPPC removed the restriction for all tephritid irradiation treatments, not only PT#33.
Response: Thank you. We revised Lines 103–108 to reflect that the restriction was removed for all relevant treatments, including PT#31.
Lines 112–113:
Reviewer: Delete “Therefore” and replace “melon fly” with “tephritids”
Response: Done. Thank you.
Section 2.3 (Statistical Method):
Reviewer: I am still not in agreement with using ANOVA with these data, but I accept your rebuttal.
Response: We sincerely appreciate your thoughtful evaluation and your acceptance of our revised analysis.
Table 1 – Larvae Number Clarification:
Reviewer: Clarify if the number of larvae treated is combined across doses and if numbers were roughly the same at each dose.
Response: Yes, the numbers shown in Table 1 represent the total number of larvae treated at each oxygen level, combining all doses. The number of larvae treated per dose was approximately equal. We appreciate your suggestion to increase the sample size at higher doses in future studies.
Lines 335–336:
Reviewer: “Data require,” not “requires.”
Response: Corrected. Thank you.
Lines 342–345:
Reviewer: Don’t over-interpret the comparison; delete the last sentence.
Response: Thank you for the suggestion. We deleted the last two sentences of the paragraph and added: “This value is within the range of previous reports.” (Lines 342–343) to maintain the integrity of the discussion.
Lines 348–349:
Reviewer: The explanation for overdosing isn’t fully accurate.
Response: Thank you for pointing this out. We revised the sentence to:
“In our understanding, this overdosing strategy is primarily intended to ensure that treatment parameters do not fall below required levels, and it may also incidentally help counteract potential radioprotective effects observed under certain modified atmospheres.”
We appreciate your insights and have updated the text accordingly.
Lack of Independent Verification (Discussion):
Reviewer: One thing you may have not addressed (adequately) in your introduction or discussion is that for irradiation, there is no independent verification of treatment efficacy as there is for all other treatments (dead insects upon inspection). Finding live insects after irradiation treatment is acceptable, so one would never know if the treatment was based on inadequate research or not applied properly. And all other major treatment categories (fumigation, cold, heat) have been found to fail at one time or another upon inspection. We will never know if irradiation fails. That is why it is important to investigate factors that might reduce efficacy, such as low oxygen. References cited: Some of the links to references link to other references or none at all. Need to check them all.
Response: Thank you for this valuable and insightful comment. We fully agree that, unlike other phytosanitary treatments such as fumigation, cold, or heat, the efficacy of irradiation cannot be independently verified through the detection of dead insects upon inspection. The acceptance of live but sterile insects presents a regulatory challenge, as it may mask cases of improper application or insufficient research. This limitation underscores the importance of proactively investigating factors—such as low oxygen—that may compromise treatment efficacy.
In response to your additional point, we have reviewed all reference links in the manuscript and corrected any mismatches or errors. We greatly appreciate your close reading and detailed feedback.
Reference #59 does not exist, What were the authors seeking when they put this in?
Response: Thank you for identifying this. We have corrected the citation by replacing it with the appropriate reference and made corresponding adjustments in both the main text and the reference list (Lines 363, 551–552).
Reviewer 2 Report
Comments and Suggestions for Authors
The manuscript has been significantly improved through revisions. It is ready for publication.
Author Response
Thank you very much for your positive evaluation and kind comments. We greatly appreciate your time and support throughout the review process.